# Optimizing the Flocculation Effect of Cationic Polyacrylamide Prepared with UV-Initiated Polymerization by Response Surface Methodology

**Chaochen Fu** [1,†]**, Zhengan Zhang** [2,*,†]**, Yuying Li** [2,*]**, Lin Li** [2]**, Hongtian Wang** [2]**, Shaobo Liu** [2]**, Xia Hua** [3]
**and Bailian Li** [2,4]

1   School of Water Conservancy and Hydroelectric Power, Hebei University of Engineering,
    Handan 056038, China; fuchaochen@126.com
2   International Joint Laboratory of Watershed Ecological Security for Water Source Region of Middle Route
    Project of South-North Water Diversion in Henan Province, School of Water Resources and Environment
    Engineering, Nanyang Normal University, Nanyang 473061, China
3   Sichuan Kelun-Biotech Biopharmaceutical Co., Ltd., Chengdu 611100, China
4   Department of Botany and Plant Sciences, University of California, Riverside, CA 92521, USA
*   Correspondence: zhangzhengan0397@163.com (Z.Z.); lyying200508@163.com (Y.L.)
†   These authors contributed equally to this work and regarded as co-first author.

**Abstract:** Cationic polyacrylamide (CPAM) is a commonly used flocculant for water treatment. Factors that affect the flocculation effect and can be controlled manually include the type and dosage of CPAM, wastewater pH, stirring time and settling time, and their reasonable setting is critical to the flocculation effect of CPAM. In this paper, the optimal flocculation conditions of a novel CPAM were studied. First, single-factor tests were conducted to preliminarily explore the optimal range of factors that influence CPAM flocculation, and then response surface methodology (RSM) tests were performed to accurately determine the optimums of the influencing factors. The results showed that the flocculation effect was better when the intrinsic viscosity was larger or the cationic degree of CPAM was higher. The CPAM dosage, wastewater pH and stirring time significantly impacted the flocculation effect, and inflection points were observed. A model that could guide CPAM-8.14-40.2 flocculation was obtained by RSM tests. The model optimization showed that the optimal flocculation conditions of CPAM-8.14-40.2 for treating wastewater prepared with kaolin were as follows: the CPAM dosage, wastewater pH and stirring time were 5.83 mg·L$^{-1}$, 7.28, and 5.95 min, respectively, and the turbidity of the treated wastewater was reduced to 6.24 NTU.

**Keywords:** cationic polyacrylamide; response surface methodology; turbidity; flocculation; intrinsic viscosity; cationic degree





## 1. Introduction

At present, flocculation, photocatalysis, biodegradation and other methods are commonly used for water treatment [1,2]. Among these methods, flocculation necessitates the use of a water purifier, namely, flocculants, which are divided into the following categories: organic flocculants, inorganic flocculants and organic–inorganic composite flocculants [3]. CPAM is a type of organic flocculant, and its flocculation mechanism mainly includes adsorption bridging, charge neutralization (including the electrostatic patch effect), catching-sweeping and a combination of these mechanisms. In particular, its adsorption bridging and charge neutralization are very strong because of its long molecular chain and positive charge group, which can prompt pollutants in sewage to aggregate into larger flocs and settle, especially negatively charged colloidal particles [4]. Most sewage has the property of negatively charged colloid; thus, flocculation treatment with CPAM is more suitable for sewage. As a result, compared to other types of flocculants, this method is more widely used. In engineering practice, CPAM, as a flocculant, is often used to dehydrate sludge

and remove suspended solids in sewage; in addition, CPAM is used to remove chemical oxygen demand (COD), total phosphorus (TP) and heavy metals from sewage. Zhengan Zhang et al. treated photosphing wastewater with CPAM and calcium chloride, and Zn, TP, and COD were reduced to 0.44, 0.33, and 38.0 mg·L$^{-1}$, respectively [5]. The flocculation of CPAM is a comprehensive process including physical, chemical and even biological actions, and its flocculation effect is affected by many factors, such as the performance of the flocculant, process design, and wastewater quality [6]. The flocculant exhibits the best effect only when each factor is set reasonably. Consequently, it is necessary to study the factors affecting the flocculation effect of CPAM and their influencing rule. Many factors affect the flocculation effect of CPAM, but those that can be controlled manually mainly include the intrinsic viscosity, cationic degree and usage amount of CPAM, the treated wastewater pH, the stirring and settling time of flocculation, etc., and their reasonable setting is critical to the flocculation effect of CPAM [7].

Most wastewater contains a pollutant, namely, suspended solids, which are usually removed from wastewater by flocculation. Generally, the suspended solids in wastewater contain a large amount of kaolin, and the physical and chemical properties of wastewater containing suspended solids are very similar to those of wastewater prepared with kaolin, such as domestic sewage and wastewater from some industries. Most sewage containing suspended solids exhibits a colloidal property and negative charge and is very suitable for flocculation treatment with CPAM.

UV-initiated polymerization is a new method for preparing organic flocculants, and there have been many studies in this field [2]. For example, Yongjun Sun et al. prepared the terpolymer of AM, DAC, butyl-acrylate by UV-initiated polymerization and researched its molecular structure [8]. Huaili Zheng et al. successfully synthesized CPAM by UV-initiated template polymerization and found the cationic microblock structure in its molecular chain [9]. However, studies on how to improve the flocculation efficiency of CPAM prepared with UV-initiated polymerization have rarely been reported. To compensate for the limitations of the present research, this paper used CPAMs prepared with UV-initiated polymerization to treat wastewater prepared with kaolin. Preliminary studies, namely, single-factor experiments, were first conducted to explore the optimal range of factors that influence CPAM. Then, RSM tests were performed to accurately determine the optimums of influence factors.

## 2. Materials and Methods

### 2.1. Materials

The monomer acrylamide (AM) was supplied by Chongqing Lanjie Tap Water Company (Chongqing, China). The cationic monomer dimethyldiallyl ammonium chloride (DMD) was obtained from Jinan Yifan Chemical Co., Ltd. (Jinan, China). The photoinitiator 2,2′-azobis (2-methylpropionamide) dihydrochloride (V-50) was purchased from Ruihong Biological Technology Co., Ltd. (Shanghai, China). Urea [CO(NH2)2] was obtained from Tianjin Kaitong Chemical Reagent Co., Ltd. (Tianjin, China). Kaolin was obtained from Tianjin Fuchen Chemical Reagent Factory. (Tianjin, China). DMD and AM were of technical grade, and other reagents, including ethanol, V-50, urea, kaolin, hydrochloric acid (HCl) and sodium hydroxide (NaOH), were of analytical grade. All aqueous and standard solutions were prepared with homemade deionized water. The purity of nitrogen gas was higher than 99.99%.

### 2.2. Polymer Preparation

The predetermined dosages of monomers (AM, 57.0 mmol; DMD, 24.4 mmol) were put in a glass reaction vessel, the predetermined dosage of distilled water was immediately added into the vessel, urea (3.0 wt ‰ of total mass) was added to increase solubility, and then the pH was adjusted to 7 by adding HCl or NaOH solution. After that, the solution was purged with nitrogen bubbling for 20 min to remove oxygen. In addition, the prearranged dosage of V-50 initiator was added into the solution. After another 10-min purge with

nitrogen, the reaction vessel was sealed, and the solution was irradiated for 80 min with a UV lamp (main wavelength 365 nm, Shanghai Jiguang Special Lighting Electric Factory, Shanghai, China) to ensure that the monomer in the reaction vessel could be polymerized as fully as possible. Then, the reaction bottle was placed at room temperature to cool and develop the polymer for 60 min. Then, the prepared polymer was completely dissolved in deionized water, and the polymer solution was adjusted to pH less than 2 and then purified with ethanol to obtain the CPAM product. During polymerization, the dosage of the photoinitiator and the molar ratio of monomer raw materials were changed to obtain a series of CPAM products with different intrinsic viscosities or cationic degrees. The possible synthesis reaction during polymerization is shown in Figure 1:

**Figure 1.** Possible reaction to synthesize CPAM.

The intrinsic viscosities of all prepared CPAM products were measured in a 1.0 mol/L NaCl solution with an Ubbelohde capillary viscometer (Shanghai Shenyi Glass Instrument Co., Ltd., Shanghai, China) at 30 ± 0.05 °C [10]. The cationic degrees of all prepared CPAM products were determined by colloid titration [11]. According to the test requirements, some of the products listed in Table 1 were selected and used as flocculants to treat wastewater prepared with kaolin.

**Table 1.** Information about the CPAM products used in the flocculation test.

| CPAM Number | Intrinsic Viscosity (dL·g$^{-1}$) | Cationic Degree (%) | The Main Parameters of the Polymerization Process | | |
| --- | --- | --- | --- | --- | --- |
| | | | Molar Ratio of AM and DMD | Dosage of V-50 Initiator (%) | Total Monomer Content (%) |
| CPAM-8.03-15.8 | 8.03 | 15.8 | 7:3 | 0.08 | 30 |
| CPAM-7.91-33.5 | 7.91 | 33.5 | 5:5 | 0.05 | 30 |
| CPAM-8.14-40.2 | 8.14 | 40.2 | 4:6 | 0.03 | 30 |
| CPAM-5.82-28.1 | 5.82 | 28.1 | 5:5 | 0.13 | 30 |
| CPAM-8.12-27.9 | 8.12 | 27.9 | 5:5 | 0.04 | 30 |
| CPAM-9.51-28.3 | 9.51 | 28.3 | 5:5 | 0.03 | 30 |

## 3. Single-Factor Flocculation Test, Results and Discussion

### 3.1. Single Factor Flocculation Test Design

To study the influences of the intrinsic viscosity and cationic degree of CPAM, wastewater pH, stirring time and settling time on the flocculation efficiency of CPAM, gradient flocculation tests were carried out to explore the optimal range of each factor and provided data for the follow-up design of the RSM test.

The kaolin reagent used in this test was of analytical purity. The crystal chemical composition of kaolin was $Al_2O_3 \cdot 2SiO_2 \cdot 2H_2O$. The particles were mostly less than 5 μm in size. After kaolin was mixed with water, wastewater was formed that contained suspended

solids, and had the property of negatively charged colloid. The wastewater used in the flocculation test was prepared with kaolin and purified water, and the concentration of kaolin in the wastewater was 2000 mg·L$^{-1}$. The original turbidity of the suspension exceeded the upper limit of the turbidity meter (HACH, Loveland, CO, USA). The test process was as follows: a beaker was used to hold 500 mL of kaolin wastewater, its pH value was adjusted to the predetermined value by adding HCl or NaOH solution, the predetermined amount of specific CPAM product was added to the wastewater, the wastewater was stirred with a ZR4-6 coagulation experiment blender (Shenzhen Zhongrunshui Industrial Technology Development Co., Ltd., Shenzhen, China) at a stirring speed of 300 rpm for the predetermined time, the solution was settled for the predetermined time, and the turbidity of the supernatant was measured with a turbidity meter. The flocculation effect of CPAM was evaluated, and the influences of the intrinsic viscosity and cationic degree of CPAM, the wastewater pH, the stirring time and the settling time on the flocculation efficiency of CPAM were analyzed according to the measurement results.

### 3.2. Results and Discussion of the Single-Factor Flocculation Test

3.2.1. Impact of Wastewater pH on the Flocculation Effect of CPAM

Figure 2 shows the test results that the CPAM numbered CPAM-8.12-27.9 treated the prepared wastewater samples, and the conditions of flocculation treatment were as follows: the dosage of CPAM, stirring time and settling time were 8 mg·L$^{-1}$, 5 min and 30 min, respectively, and the pH values of wastewater samples were adjusted according to the predetermined gradient values.

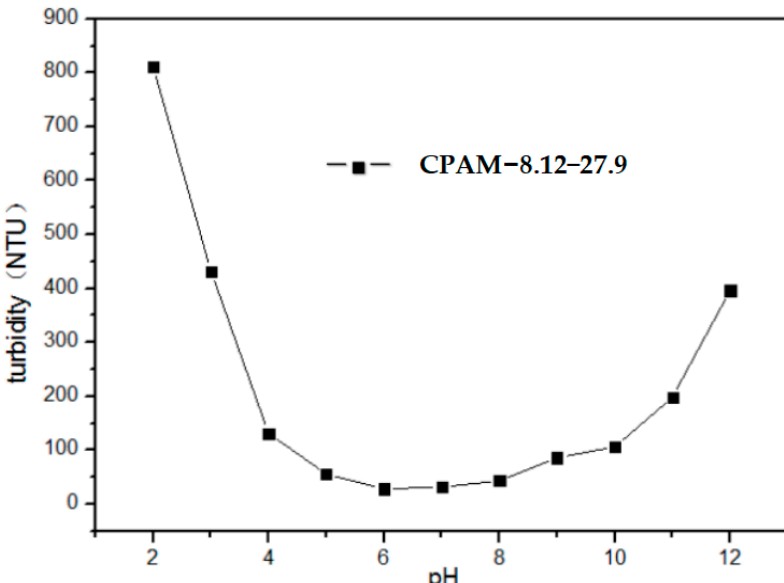

**Figure 2.** Impact of wastewater pH on the flocculation effect of CPAM.

Figure 2 shows that the wastewater turbidity dropped sharply in pH values less than 4, dropped slowly when the pH was between 4 and 6, remained basically stable when the pH was between 6 and 8, and rose significantly when the pH exceeded 8. The minimum turbidity was 28.7 NTU, and the corresponding wastewater pH was 6. The test results indicated that the wastewater pH exhibited a great influence on the flocculation of CPAM, and the pH range suitable for CPAM flocculation was 5 to 8.

The test results indicated that the flocculation effect of CPAM was very poor when the wastewater pH was less than 5. The main reasons for this phenomenon were as follows: First, the negative charges carried by colloidal particles were neutralized by hydrogen ions, and the colloidal particles became electrically neutral or even positively charged and were difficult to coagulate [12,13]. In addition, when CPAM encounters positively charged groups, its charge neutralization advantage cannot be fully exploited, which aggravates the

electrostatic repulsion between colloidal particles [14]. The decrease in sewage turbidity was mainly due to the adsorption bridging of CPAM.

The test results also showed that the flocculation performance of CPAM deteriorated when the wastewater pH exceeded 8. The main reason was that a large number of hydroxyl ions in wastewater neutralized the positive charge of CPAM and weakened its charge neutralization [5,15].

### 3.2.2. Impacts of the Dosage and Intrinsic Viscosity of CPAM on Its Flocculation Effect

Figure 3 shows the test results of treating the prepared wastewater samples with CPAM products, and their numbers were CPAM-9.51-28.3, CPAM-8.12-27.9 and CPAM-5.82-28.1; the products had nearly the same cationic degree and significantly different intrinsic viscosities. The conditions of flocculation treatment were identical to those in Section 3.2.1 except that the wastewater pH was adjusted to 6, and the dosages of each CPAM were added according to the predetermined gradient dosages.

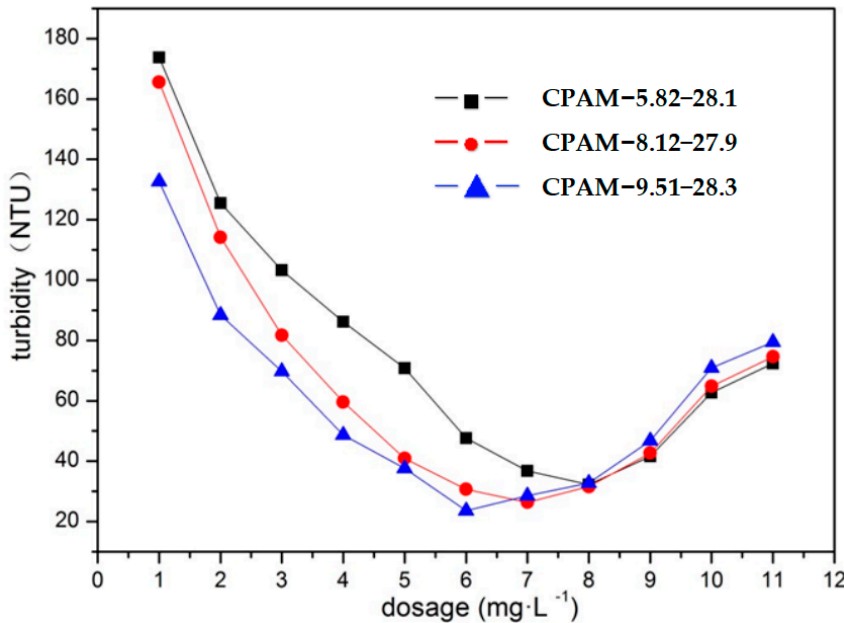

**Figure 3.** Impact of the dosage and intrinsic viscosity of CPAM on its flocculation effect.

CPAM contains long molecular chains and many cationic units, so its main flocculation methods are adsorption bridging and charge neutralization [16]. In general, the larger the molecular weight of CPAM is, the longer its molecular chain, the stronger its adsorption bridging and the better its flocculation performance; the higher the cationic degree of CPAM is, the stronger its charge neutralization [8]. The molecular weight of the polymer is converted according to the intrinsic viscosity of the polymer, and the two are generally positively correlated [17]. Therefore, the flocculation effect and main flocculation mechanism of CPAM can be judged by its intrinsic viscosity and cationic degree. Figure 3 shows that the wastewater turbidities decreased rapidly at first and then gradually increased with increasing CPAM dosage. Thus, excessive CPAM dosage was not conducive to flocculation because adding excessive flocculants made the coagulated flocs positively charged, and the flocs rediffused and dissolved in the wastewater because of their electrostatic repulsion; this phenomenon was also called floc restabilization [18]. The minimum turbidities obtained by treating wastewater with CPAM-5.82-28.1, CPAM-8.12-27.9, and CPAM-9.51-28.3 were 32.2, 26.4 and 23.6 NTU, respectively, and the corresponding dosages were 8, 7 and 6 mg·L$^{-1}$, respectively. When comparing the variation trend for wastewater turbidity, it was found that the flocculation effect of the CPAM products from good to poor was in the order of CPAM-9.51-28.3, CPAM-8.12-27.9 and CPAM-5.82-28.1. The flocculation test conditions

and treated wastewater were identical, and the three CPAM products had almost the same charge neutralization effect because of their nearly equal cationic degree. Therefore, the only reason for the different flocculation results was their different intrinsic viscosities [9]. Normally, the greater the intrinsic viscosity of CPAM is, the stronger its adsorption bridging, and the lower the turbidity wastewater treated, which was also confirmed by the results shown in Figure 3.

### 3.2.3. Impact of the Dosage and Cationic Degree of CPAM on Its Flocculation Effect

Figure 4 shows the test results of treating the prepared wastewater with three different CPAM products. Their numbers were CPAM-8.03-15.8, CPAM-7.91-33.5, and CPAM-8.14-40.2, and they had almost the same intrinsic viscosity and significantly different cationic degrees. The conditions of flocculation treatment were the same as those in Section 3.2.2.

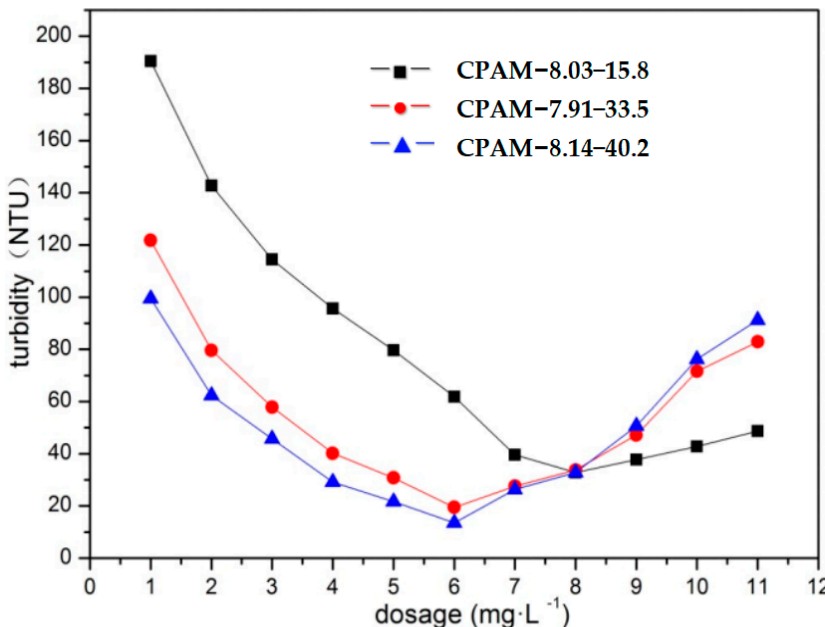

**Figure 4.** Impact of dosage and cationic degree of CPAM on its flocculation effect.

Figure 4 shows that the wastewater turbidity caused by the three CPAMs showed a similar trend; that is, with the increase in the dosage of CPAM, the turbidity of the wastewater first decreased to the minimum value and then gradually increased. This phenomenon once again proved that excessive CPAM dosage was not conducive to flocculation. When comparing the flocculation efficiency of the three CPAMs, it was found that their flocculation efficiencies were significantly different. The flocculation efficiency of CPAM-8.14-40.2 was the best, and when its dosage was 6 mg·L$^{-1}$, the wastewater turbidity decreased to the lowest value (13.6 NTU). That of CPAM-8.03-15.8 was the worst, and when its dosage was 8 mg·L$^{-1}$, the wastewater turbidity decreased to the lowest value; however, it still reached 32.9 NTU. These differences were mainly caused by the different cationic degrees of the three CPAMs. Normally, the greater the cationic degree of CPAM is, the stronger the charge neutralization, and the lower the turbidity wastewater treated [19], which was also confirmed by the test results. Figure 3 also shows that the three CPAM products led to colloid destabilization; the most obvious was CPAM-8.14-40.2 with the highest cationic degree, and the least obvious was CPAM-8.03-15.8 with the lowest cationic degree, which indicated that the higher the cationic degree of CPAM is, the easier it is to destabilize the colloid. Therefore, for CPAM with a high cationic degree, it is critical to add an appropriate dosage during its flocculation process.

### 3.2.4. Impact of Stirring Time on Flocculation Properties of CPAM

Figure 5 shows the test results of treating the prepared wastewater with CPAM numbered CPAM-8.14-40.2. The conditions of flocculation treatment were the same as those in Section 3.2.2 except that the dosage of CPAM was 6 mg·L$^{-1}$, and the stirring times were set according to the predetermined gradient times.

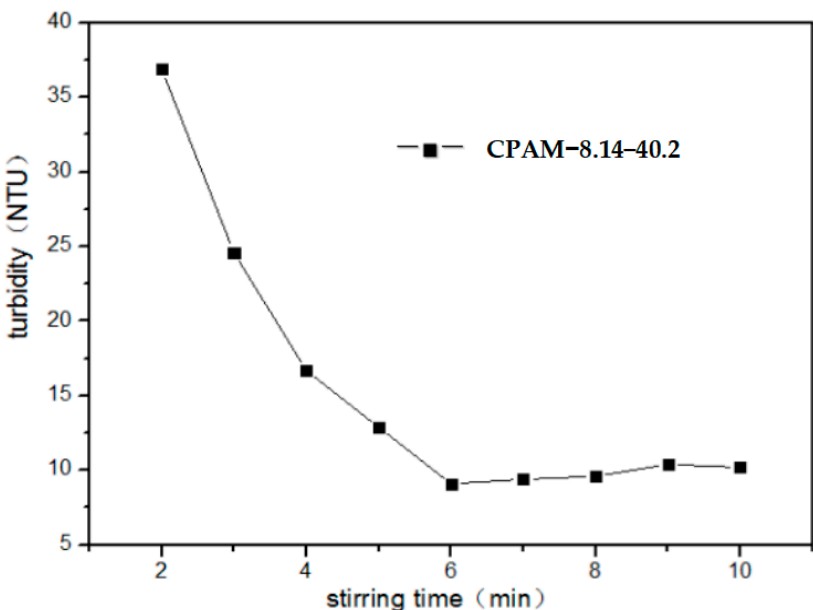

**Figure 5.** Impact of the stirring time on CPAM flocculation performance.

As shown in Figure 5, with the extension of stirring time, the turbidity of wastewater rapidly decreased to the lowest value (9.1 NTU) at first and then slightly rose. In the first 6 min of stirring time, the CPAM and sewage were completely mixed, the flocculation reaction finished, and flocs were formed with good settling performance; therefore, stopping stirring at that moment was the most conducive to floc settlement, and the corresponding wastewater turbidity was the lowest. If the stirring time was too long, the large formed flocs were broken into small flocs with poor settlement performance and eventually lead to a decline in flocculation efficiency [18,20], which was why the turbidity of the wastewater increased slightly after the stirring time exceeded 6 min.

### 3.2.5. Impact of the Settling Time on CPAM Flocculation Efficiency

Figure 6 shows the test results obtained from treating the prepared wastewater with the CPAM product numbered CPAM-8.14-40.2. The conditions of flocculation treatment were the same as those in Section 3.2.4 except that the stirring time was 6 min, and the settling times were set according to the predetermined gradient times. As shown in Figure 5, the turbidity change in wastewater underwent three stages. The first was the rapid sedimentation stage, and its time range was 0~10 min. During this stage, the suspended solids in the wastewater first coagulated into small flocs under the electric neutralization of CPAM, and then these small flocs were connected in series to form large flocs under the adsorption bridging of CPAM. These large flocs sank to the beaker bottom under gravity [21]. The second stage was the slow settling stage, with a time range of 10–25 min. In this stage, some small and light flocs remained in the wastewater. Due to their low gravity and high buoyancy, they settled very slowly and took a long time to sink into the bottom of the beaker [22]. Therefore, the turbidity of wastewater in this stage continued to slowly decrease to the minimum value (8.6 NTU). The third stage was the stabilization stage after 25 min. In this stage, very small flocs and suspended solids remained in the wastewater, but they were difficult or unable to settle, so the wastewater turbidity did not

change much and basically remained stable at approximately 8.6 NTU. Therefore, at least 25 min of settling time was necessary to achieve a better flocculation effect.

After the flocculation test, the beaker containing the treated sewage was stored for many days, but the sediment did not float up, which was different from the phenomenon that the sediment in the secondary sedimentation tank of the sewage activated sludge process floats up if the sludge retention time is too long [23]. A basic explanation is that the chemical compositions of the two sediments are different. The chemical composition of the precipitate in this experiment was mainly silicon dioxide, silicate and other inorganic substances, which did not react and led to a change in the phase state [24,25]. However, the sediments in the secondary sedimentation tank of the sewage activated sludge process are mainly composed of biodegradable organic matter and microorganisms. If the sediment stays in the secondary sedimentation tank for too long, it degrades and releases biogas, which causes the sediment to float up [23]. In addition, only a short time was needed for the precipitation process to complete in this test because the sediments were mainly inorganic substances with high density and exhibited good settling performance.

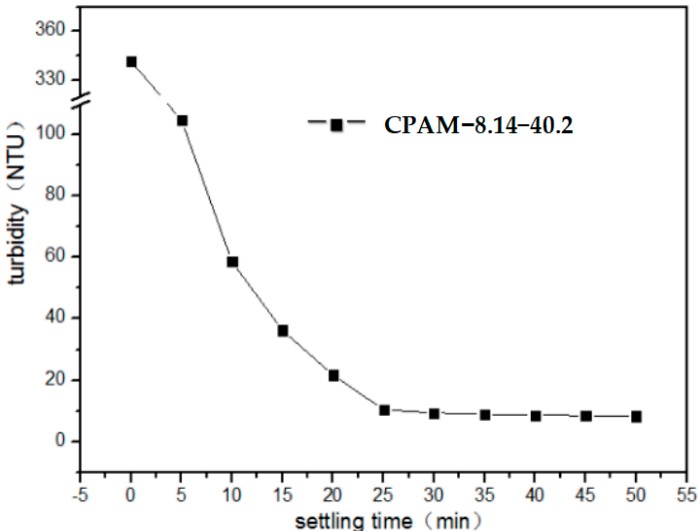

**Figure 6.** Impact of the settling time on the flocculation performance of CPAM.

## 4. RSM Flocculation Test, Results and Discussion

### 4.1. RSM Flocculation Test Design

A single factor test generally has the following two shortcomings: one is that the test ignores the influence of the interaction between variable factors on the test results; the other is that the span of the variable value is too large to accurately capture the optimal value of the variable [26,27]. To obtain the best variable level and test result, further optimization of test design, such as orthogonal design or RSM, should be carried out according to the single factor test results [28]. RSM is an excellent method for optimizing and verifying scientific research and industrial studies. It uses the multivariate quadratic regression equation to fit the relationship between the index and influencing factors through regression equation analysis. This method aims to find the best process parameters and has the ability to provide maximum information with minimum experiments. Compared with orthogonal design, it is more intuitive and easier to reflect the optimal value of dependent variables [29]. The Box–Behnken design (BBD), a kind of RSM design, is the most frequently used design in pioneering studies because it is more scientific compared with other designs in RSM [28,30,31].

The results of single-factor tests showed that the CPAM dosage, wastewater pH, stirring time and settling time had significant impacts on the flocculation efficiency of CPAM. The intrinsic viscosity, cationic degree and settling time were positively correlated with the flocculation efficiency of CPAM, and their optimums were those obtained by

the single factor test. However, the factors including CPAM dosage, wastewater pH and stirring time had both positive and negative correlations with the flocculation efficiency of CPAM; thus, there were inflection points, and the variable values corresponding to inflection points were not necessarily the optimum but were close. Hence, on the basis of the single-factor test results, a BBD test was designed and conducted. The CPAM product No. CPAM-8.14-40.2 in Table 1 was selected as the flocculant, the wastewater identical to that of the single-factor test was the treated object, reducing the wastewater turbidity was the optimization goal, and the CPAM dosage, the wastewater pH and the stirring time were influencing factors. The BBD test was designed with Design-Expert 8.0.6 software [18], and each influence factor was set at three experimental levels, namely, high (+1), low (−1), and central point (basic level 0). The experimental levels and corresponding values of independent variables were determined and are listed in Table 2. The settling times of all flocculation tests were the same, i.e., 30 min.

**Table 2.** Experimental levels of independent test variables.

| Variable Code | Variables | Variable Levels and Corresponding Values | | |
| | | −1 | 0 | 1 |
| --- | --- | --- | --- | --- |
| $Z_1$ | CPAM dosage (mg·L$^{-1}$) | 5 | 6 | 7 |
| $Z_2$ | Wastewater pH | 4 | 6 | 8 |
| $Z_3$ | Stirring time (minutes) | 4 | 6 | 8 |

*4.2. Results and Discussion of the RSM Flocculation Test*

4.2.1. Discussion of RSM Test Results

According to the scheme of the BBD test, a total of 17 groups of flocculation tests were carried out, including 12 factorial tests and five central tests for inspection errors. The variable values and the corresponding turbidity of each flocculating test are listed in Table 3. The results showed that the turbidities of the five central tests were significantly lower than those of the others, which was consistent with the results of the single factor test and proved that the central point values of the influencing factors of the RSM tests were reasonable but not necessarily the optimal values, which could be obtained through further RSM analysis.

**Table 3.** The actual response values and predicted response values of BDD tests.

| Run | CPAM Dosage (mg·L$^{-1}$) | Wastewater pH | Stirring Time (Minutes) | Response Value of Turbidity (NTU) | | |
| | | | | Actual | Predicted | |
| | | | | | Equation (2) | Equation (3) |
| --- | --- | --- | --- | --- | --- | --- |
| 1 | 6.0 | 6.0 | 6.0 | 8.90 | 8.82 | 8.82 |
| 2 | 6.0 | 4.0 | 8.0 | 32.80 | 32.45 | 32.45 |
| 3 | 7.0 | 4.0 | 6.0 | 33.30 | 33.50 | 32.43 |
| 4 | 6.0 | 6.0 | 6.0 | 8.80 | 8.82 | 8.82 |
| 5 | 6.0 | 4.0 | 4.0 | 33.10 | 32.85 | 32.85 |
| 6 | 6.0 | 6.0 | 6.0 | 8.10 | 8.82 | 8.82 |
| 7 | 6.0 | 8.0 | 4.0 | 16.50 | 16.85 | 16.85 |
| 8 | 5.0 | 8.0 | 6.0 | 13.00 | 12.80 | 12.73 |
| 9 | 5.0 | 6.0 | 8.0 | 24.30 | 24.25 | 24.25 |
| 10 | 5.0 | 6.0 | 4.0 | 25.80 | 26.65 | 25.65 |
| 11 | 6.0 | 6.0 | 6.0 | 9.20 | 8.82 | 8.82 |
| 12 | 5.0 | 4.0 | 6.0 | 27.10 | 27.50 | 27.58 |
| 13 | 7.0 | 6.0 | 8.0 | 32.10 | 32.25 | 32.25 |
| 14 | 7.0 | 8.0 | 6.0 | 18.9 | 18.50 | 18.58 |
| 15 | 6.0 | 6.0 | 6.0 | 9.10 | 8.82 | 8.82 |
| 16 | 7.0 | 6.0 | 4.0 | 29.3 | 29.35 | 29.35 |
| 17 | 6.0 | 8.0 | 8.0 | 18.5 | 18.75 | 18.75 |

4.2.2. Model Fitting

In terms of linear, quadratic, and cross terms, the quadratic equation Model (Y) was constructed according to Equation (1) as follows [32,33]:

$$Y = A_0 + A_1Z_1 + A_2Z_2 + A_3Z_3 + A_{12}Z_{12} + A_{13}Z_{13} + A_{23}Z_{23} + A_{11}Z_1^2 + A_{22}Z_2^2 + A_{33}Z_3^2, \tag{1}$$

Equation (1) reflects the relationship between variables and response. In this study, Y referred to the response to be modeled, i.e., wastewater turbidity (NTU); $Z_1$, $Z_2$ and $Z_3$ refer to the first-order terms of variables, i.e., the CPAM dosage (mg·L$^{-1}$), the wastewater pH and the stirring time (minutes), respectively; $Z_1^2$, $Z_2^2$ and $Z_3^2$ refer to their quadratic terms; and $Z_{12}$, $Z_{13}$ and $Z_{23}$ refer to the corresponding terms of interaction effects between two variables, respectively. $A_0$ was a constant term; $A_1$, $A_2$ and $A_3$ refer to the primary linear coefficients of the CPAM dosage (mg·L$^{-1}$), the wastewater pH and the stirring time (minutes), respectively; $A_{11}$, $A_{22}$ and $A_{33}$ represent their secondary term coefficients, respectively; and $A_{12}$, $A_{13}$ and $A_{23}$ represent the interaction term coefficients among variables, respectively.

According to the response results of the model, analysis of variance (ANOVA) was applied to analyze the feasibility of establishing the quadratic equation model between the variables and the responses [34]. To check the statistical significance of the quadratic equation model and test variables, F tests and *p* values at the 95% confidence level were used. The modeling quality of the model was tested based on the coefficient of determination $R^2$ and adjusted $R^2$. Additionally, the interaction effects of the factors ($Z_{12}$, $Z_{13}$ and $Z_{23}$) on the response value were analyzed using three-dimensional plots and two-dimensional contour graphs [35].

Using the data in Table 3, regression simulation was conducted according to Equation (1), the ternary quadratic polynomial regression model between response and variables was obtained, and the final equation in terms of actual factors is shown in Equation (2) as follows:

$$Y = 8.820 + 2.925Z_1 - 7.425Z_2 + 0.375Z_3 - 0.075Z_{12} + 1.075Z_{13} + 0.575Z_{23} + 8.453Z_1^2 + 5.803Z_2^2 + 10.603Z_3^2 \tag{2}$$

The ANOVA for response surface quadratic model, i.e., Equation (2), was conducted, the significance of the influence of each variable was tested, and the results are listed in Table 4. Generally, "*p* values Prob > F" less than 0.0500 indicate that the model terms are significant [26,35] In this case, $Z_1$, $Z_2$, $Z_{13}$, $Z_{23}$, $Z_1^2$, $Z_2^2$, and $Z_3^2$ were all significant model terms and had significant impacts on wastewater turbidity. The "*p* values Prob > F" of the model were less than 0.0500, which implied that the model was significant. The "Lack of Fit F value" of 1.46 implied that the lack of fit was not significant relative to the pure error and indicated that the equation was reliable [30,32]. The "Pred R-Squared" of 0.9907 was in reasonable agreement with the "Adj R-Squared" of 0.9977, which indicated that Equation (2) was well fitted and could be used to predict the turbidity of wastewater flocculated with CPAM-8.14-40.2. The predicted turbidity values of all flocculating tests are listed in Table 3.

The ANOVA results showed that Equation (2) exhibited a good fitting effect, but it also showed a minor defect, that is, the "*p* values Prob > F" of $Z_3$ and $Z_{12}$ were both greater than 0.0500, which implied that the stirring time and the interaction between CPAM dosage and wastewater pH both showed insignificant impacts on the wastewater turbidity. Therefore, the model, Equation (2), could be further improved by removing the intercepts of insignificant terms from the coded model, but only $Z_{12}$ can be removed, not $Z_3$, because $Z_{13}$, $Z_{23}$, and $Z_3^2$ exhibited significant impacts on the results of the flocculation tests. After optimization, a better fitting model was obtained, and its final equation in terms of actual factors is shown in Equation (3) as follows:

$$Y = 8.820 + 2.925Z_1 - 7.425Z_2 + 0.375Z_3 + 1.075Z_{13} + 0.575Z_{23} + 8.453Z_1^2 + 5.803Z_2^2 + 10.603Z_3^2 \tag{3}$$

**Table 4.** ANOVAs for the response surface of Equations (2) and (3).

| Source | | Sum of Squares | Df | Mean Squares | F Value | *p* Value Prob > F | Remark |
|---|---|---|---|---|---|---|---|
| Model | Equation (2) | 1532.076706 | 9 | 170.2307451 | 759.9586835 | <0.0001 | significant |
| | Equation (3) | 1532.054206 | 8 | 191.5067757 | 963.2531945 | <0.0001 | significant |
| $Z_1$-the CPAM dosage(mg·L$^{-1}$) | Equation (2) | 68.445 | 1 | 68.445 | 305.5580357 | <0.0001 | |
| | Equation (3) | 68.445 | 1 | 68.445 | 344.2690978 | <0.0001 | |
| $Z_2$-the wastewater pH | Equation (2) | 441.045 | 1 | 441.045 | 1968.950893 | <0.0001 | |
| | Equation (3) | 441.045 | 1 | 441.045 | 2218.396731 | <0.0001 | |
| $Z_3$-the stirring time(minutes) | Equation (2) | 1.125 | 1 | 1.125 | 5.022321429 | 0.06 | |
| | Equation (3) | 1.125 | 1 | 1.125 | 5.658597925 | 0.0446 | |
| $Z_{12}$ | Equation (2) | 0.0225 | 1 | 0.0225 | 0.100446429 | 0.7605 | |
| | Equation (3) | – | – | – | – | – | |
| $Z_{13}$ | Equation (2) | 4.6225 | 1 | 4.6225 | 20.63616071 | 0.0027 | |
| | Equation (3) | 4.6225 | 1 | 4.6225 | 23.25055014 | 0.0013 | |
| $Z_{23}$ | Equation (2) | 1.3225 | 1 | 1.3225 | 5.904017857 | 0.0454 | |
| | Equation (3) | 1.3225 | 1 | 1.3225 | 6.651996228 | 0.0327 | |
| $Z_1{}^2$ | Equation (2) | 300.8200263 | 1 | 300.8200263 | 1342.946546 | <0.0001 | |
| | Equation (3) | 300.8200263 | 1 | 300.8200263 | 1513.084068 | <0.0001 | |
| $Z_2{}^2$ | Equation (2) | 141.7642368 | 1 | 141.7642368 | 632.8760573 | <0.0001 | |
| | Equation (3) | 141.7642368 | 1 | 141.7642368 | 713.054948 | <0.0001 | |
| $Z_3{}^2$ | Equation (2) | 473.3179211 | 1 | 473.3179211 | 2113.026433 | <0.0001 | |
| | Equation (3) | 473.3179211 | 1 | 473.3179211 | 2380.725161 | <0.0001 | |
| Residual | Equation (2) | 1.568 | 7 | 0.224 | | | |
| | Equation (3) | 1.5905 | 8 | 0.1988125 | | | |
| Lack of fit | Equation (2) | 0.82 | 3 | 0.273333333 | 1.461675579 | 0.3512 | not significant |
| | Equation (3) | 0.8425 | 4 | 0.210625 | 1.126336898 | 0.4555 | not significant |
| Pure error | Equation (2) | 0.748 | 4 | 0.187 | | | |
| | Equation (3) | 0.748 | 4 | 0.187 | | | |
| Cor total | Equation (2) | 1533.644706 | 16 | | | | |
| | Equation (3) | 1533.644706 | 16 | | | | |
| $R^2$ | Equation (2) | 0.9938 | | | | | |
| | Equation (3) | 0.9907 | | | | | |
| $R^2{}_{adj}$ | Equation (2) | 0.9979 | | | | | |
| | Equation (3) | 0.9977 | | | | | |

The ANOVA for Response Surface Quadratic Model, i.e., Equation (3), was also conducted, and the significance of variable influence was tested. The results listed in Table 4 show that the "Lack of Fit F value" of the model, i.e., Equation (3), was approximately 1.126334, which was less than that of Equation (2) and implied that the former was more reliable. Therefore, Equation (3) was chosen to predict the turbidity of wastewater flocculated with CPAM-8.14-40.2, and the predicted values are listed in Table 3.

4.2.3. Response Surface Analysis

Equation (3) was selected to perform ANOVA to determine the impacts of interactions between variables on the flocculation effect. The result showed that the interaction between the CPAM dosage and wastewater pH had an insignificant impact on the wastewater turbidity; therefore, the interaction did not have to be analyzed in contrast to the interaction between CPAM dosage and stirring time and the interaction between wastewater pH and

stirring time. Design Expert 8.0.6 software was used to draw the response surface diagrams, which are shown in Figures 7 and 8. The influence of each factor on the wastewater turbidity could be judged by the steepness of the three-dimensional response surface. The steeper the slope of the response surface is, the more significant the influence of this factor on the test results [36,37]. The influence of the interaction between factors on the wastewater turbidity could be judged by the shape of the two-dimensional contour graph. If the contour graph was oval, it indicated that the interaction effect of the corresponding factors had a significant influence on the wastewater turbidity; however, when the contour tended to be circular, the influence was small [37–39]. One of the factors was fixed, and the influences of the other two factors on the response value were investigated. Figure 7 shows that with increasing CPAM dosage and stirring time, the wastewater turbidity first decreased and then increased under the condition of fixed wastewater pH, and when the CPAM dosage and stirring time were in the range of 5.5 to 6 mg·L$^{-1}$ and 5 to 7 min, respectively, the wastewater turbidity had a minimum value. Similarly, as shown in Figure 8, when the CPAM dosage was fixed, with increasing wastewater pH and stirring time, the wastewater turbidity showed a trend of first increasing and then decreasing; when the wastewater pH and stirring time were in the range of 6 to 8 and 5 to 7 min, respectively, the wastewater turbidity had a minimal value.

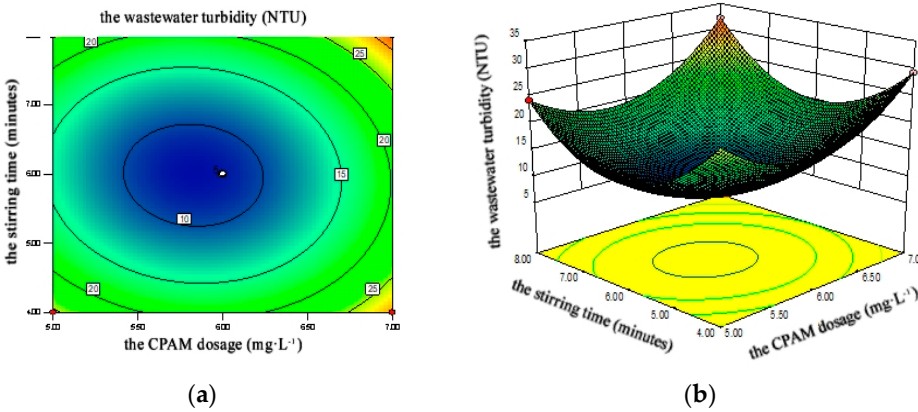

**Figure 7.** The impact of the interaction between CPAM dosage and stirring time on wastewater turbidity. (**a**) Contour diagram; (**b**) 3D surface diagram.

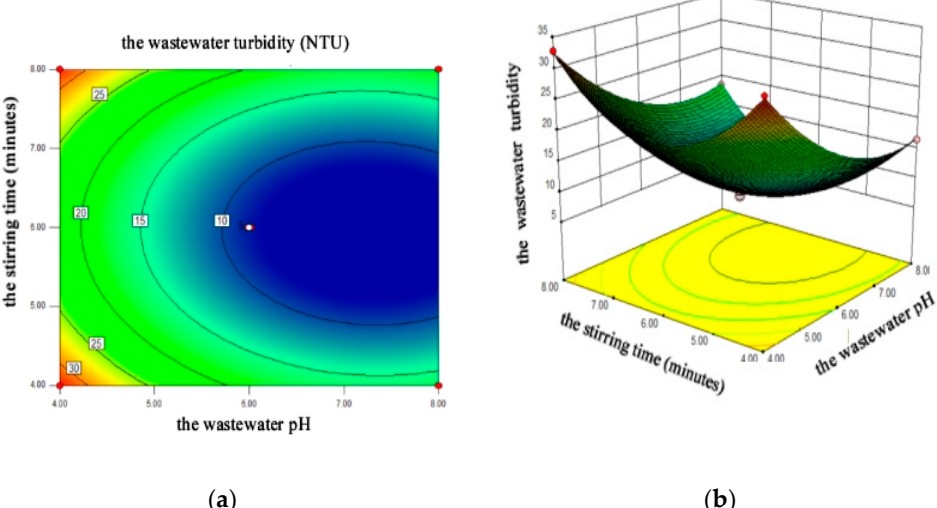

**Figure 8.** Impact of the interaction between stirring time and wastewater pH on wastewater turbidity. (**a**) Contour diagram; (**b**) 3D surface diagram.

4.2.4. Flocculating Optimization and Model Validation

In Equation (3), the first-order partial derivative was obtained and solved by being set to zero [30,39,40], and the optimal flocculating conditions were obtained as follows: the CPAM dosage, the wastewater pH and the stirring time were 5.83 mg·L$^{-1}$, 7.28, and 5.95 min, respectively, and the predicted turbidity of the treated wastewater was 6.18 NTU. To confirm the reliability of the prediction model, two runs of additional experiments were conducted under the flocculation conditions obtained from the model optimization, and the settling time was 30 min. The experimental results are listed in Table 5 and show that the average of the measured turbidities was 6.24 NTU, which is very close to the predicted value of 6.18 NTU. The error between the measured turbidity and the predicted turbidity was only 3.4%, which indicated that the prediction model could be used to guide the flocculation of CPAM [34,41].

**Table 5.** Measured and predicted values of wastewater turbidity.

| Flocculation Conditions | | | | Wastewater Turbidity (NTU) | |
|---|---|---|---|---|---|
| CPAM Dosage (mg·L$^{-1}$) | Wastewater pH | Stirring Time (Minutes) | Settling Time (Minutes) | Average of Measured Value | Predicted Value |
| 5.83 | 7.28 | 5.95 | 30 | 6.49 | 6.18 |

**5. Conclusions**

To study the optimal flocculation conditions of CPAM prepared with UV-initiated polymerization, first, single-factor tests were conducted to preliminarily explore the optimal range of influencing factors of CPAM flocculation, and then RSM tests were performed to accurately determine the optimums of influencing factors. The single-factor test results showed that the flocculation effect was better when the intrinsic viscosity was larger or the cationic degree of CPAM was higher; the CPAM dosage, wastewater pH and stirring time had significant impacts on the flocculation effect of CPAM and existed optimums. A model that could guide CPAM flocculation was obtained by RSM tests. The results of model optimization showed that the optimal flocculation conditions for treating the wastewater prepared with kaolin by CPAM-8.14-40.2 were as follows: the CPAM dosage, the wastewater pH and the stirring time were 5.83 mg·L$^{-1}$, 7.28, and 5.95 min, respectively, and the turbidity of treated wastewater was reduced to 6.24 NTU.

**Author Contributions:** Conceptualization, Z.Z., C.F. and Y.L.; methodology, Z.Z. and Y.L.; software, Z.Z. and C.F.; validation, Z.Z., Y.L. and C.F.; formal analysis, Y.L.; investigation, L.L. and S.L.; resources, L.L., B.L. and H.W.; data curation, L.L. and X.H.; writing—original draft preparation, Z.Z.; writing—review and editing, Z.Z., B.L. and Y.L.; visualization, Z.Z. and C.F.; supervision, L.L., S.L. and H.W.; project administration, Y.L.; funding acquisition, Y.L., Z.Z. and C.F. All authors have read and agreed to the published version of the manuscript.

**Funding:** This study was funded by the Key Research and Development Projects of Henan Province (No. 221111520600) and the Higher Discipline Innovation and Talent Introduction Base of Henan Province (No. CXJD2019001).

**Data Availability Statement:** Not applicable.

**Conflicts of Interest:** The authors declare no conflict of interest.

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
