# Peer review of "Optimizing the Flocculation Effect of Cationic Polyacrylamide Prepared with UV-Initiated Polymerization by Response Surface Methodology"

_water, doi:10.3390/w15061200_

Round 1
Reviewer 1 Report
The authors aimed to study some major factors, including pH value, dosage, and viscosity and cationic degree, affecting the flocculation process, and the optimal parameters were obtained based on the model optimization and experimental validation. The paper is well organized and the findings are interesting. However, there still exist some major problems as follows.
1. The authors failed to present the detailed information about Kaolin. Why choose Kaolin in this study? Does the size of Kaolin affect the experiment results?
2. In the section of Results and Discussion, the title for 3.2.2 is not correct. Check carefully and correct it.
3. Does the addition of HCl or NaOH induce the reaction between the additives and Kaolin? Does this addition affect the stability of colloids or the turbidity of the system?
4. There exist some grammar mistakes in this paper. Check throughout this paper carefully.
Reviewer 2 Report
In this work, CPAM is used as a flocculant for treatment of synthetic wastewater. The study is systematic in the field of wastewater treatment. RSM is used to predict the optimum condition for the flocculation process with good matching with experimental results. The findings are promising, however; several drawbacks affect the significance of this work.
1. Concise literature review regarding the utilization of CPAM as a flocculant should be covered.
2. Section titles should be checked, some titles are repeated in different sections, e.g: 3.2.2 and 3.2.3.
3. The methods that used to prepare CPAM with different viscosity are not clear, a table contains the proportions and/or the amount of monomer, initiator and other reactants can be utilized to summarize the preparation process.
4. The authors extensively used very long yet confused sentences, please use concise and short sentences. Thus, results' discussion is suggested to be rewritten.
5. Extensive English language editing is a MUST!
6. Abbreviations should be defined at the first appearance after full words, then it should be used alone.
Reviewer 3 Report
Title
The title of the manuscript is OK.
Graphical abstract & abstract
The abstract is OK and acceptable, but the author must add its graphical abstract to enhance its citations and downloads, which attract the readers' attention.
Introduction
The coherence is satisfactory, but the material added needs amendments. There is addition of materials required in the introduction part. There are no proper spacing given in the references while at many places grammatical errors was found that must be removed using some sort of software (Grammarly). The author of the manuscript can take help using the below DOIs that play a significant role in enhancing the quality of the introduction.
ii) 10.19080/JOJMS.2022.07.555703
ii) 10.1016/j.chemosphere.2022.136982
iii) 10.1016/j.saa.2021.119644
If the author used these DOIs in the manuscript, follow the instructions mentioned above and ignore the DOIs.
Material and methods (section 3)
In the section 3.2.2 and 3.2.3, (the conditions of flocculation treatment were roughly as follows: the wastewater pH, stirring time and settling time were 6, 5 minutes and 30 minutes respectively, and the dosages of each CPAM were 204 added according to the predetermined gradient dosages) the previous lines were repeated, the author of the manuscript should write this sentence only in section 3.2.2 while for the later section only different point is be mentioned while other stuff like figures and the information provided is smooth and acceptable.
Results and discussion
This section is divided, one part is written in 3rd section while the other part is written in 4th section. In the section 4, whole processes and RSM should be discussed while in section 3 flocculation, materials and methods are enough. The author should manage the names of the both sections. No doubt work and results are satisfactory.
Conclusion
The conclusion of the manuscript is satisfactory.
Views of the reviewer
the intrinsic viscosity, cationic degree and settling time of CPAM, figures, tables, material added, RSM were well managed and elaborated but there are some grammatical errors present in the manuscript that can be ignored, references in the manuscript are not placed with proper spacing, repetition of lines in couple of sections and the results and discussion section is splitted into couple of parts it should be placed in a proper sequence so that it could be understand easily. overall the manuscript is ok. The author needs to overcome the above mentioned minor revisions.
Reviewer 4 Report
The authors first investigate single factor correlation of dosage, wastewater pH, stirring time, settling time, etc. to flocculation. Then a response surface quadratic model is fitted for impact of interaction. According to this report, the model is able to predict pretty accurate wastewater turbidity for a set of flocculation conditions. All data is demonstrated clearly with tables and figures. Overall it is a well-organized research article relevant to the journal's scope.
Round 2
Reviewer 2 Report
Since the authors outlined most previous comments, I thank the authors for their efforts. However, few comments are observed as bellow:
1. English language needs further editing.
2. The novelty point of this manuscript should be highlighted distinctly.